# A Spectral Investigation of Criticality and Crossover Effects in Two and Three Dimensions: Short Timescales with Small Systems in Minute Random Matrices

**DOI:** 10.3390/e26050395

**Published:** 2024-04-30

**Authors:** Eliseu Venites Filho, Roberto da Silva, José Roberto Drugowich de Felício

**Affiliations:** 1Instituto de Física, Universidade Federal do Rio Grande do Sul, Av. Bento Gonçalves, 9500, Porto Alegre CEP 91501-970, RS, Brazil; eliseuv816@gmail.com; 2Departamento de Física, Faculdade de Filosofia Ciências e Letras de Ribeirão Preto, Universidade de São Paulo, Av. dos Bandeirantes 3900, Ribeirão Preto CEP 14040-905, SP, Brazil; drugo@usp.br

**Keywords:** random matrices, Wishart matrix, phase transitions, crossover phenomena

## Abstract

Random matrix theory, particularly using matrices akin to the Wishart ensemble, has proven successful in elucidating the thermodynamic characteristics of critical behavior in spin systems across varying interaction ranges. This paper explores the applicability of such methods in investigating critical phenomena and the crossover to tricritical points within the Blume–Capel model. Through an analysis of eigenvalue mean, dispersion, and extrema statistics, we demonstrate the efficacy of these spectral techniques in characterizing critical points in both two and three dimensions. Crucially, we propose a significant modification to this spectral approach, which emerges as a versatile tool for studying critical phenomena. Unlike traditional methods that eschew diagonalization, our method excels in handling short timescales and small system sizes, widening the scope of inquiry into critical behavior.

## 1. Introduction

The phenomenology of critical phenomena, encompassing phase transitions in diverse contexts, stands as a cornerstone within the framework of Statistical Mechanics theory. Initially conceived within the realm of many-body physics, it has evolved into a concept with far-reaching applications spanning disciplines such as economics, network theory, sociophysics, game theory, and numerous others [1,2,3,4,5].

In spin systems, a particularly effective approach to delve into critical phenomena involves conducting short-time dynamics studies. This method entails preparing systems with carefully chosen initial conditions and then analyzing temporal averages. Through this analysis, the manifestation of critical behavior in the system is revealed via power-law dynamics [6,7,8,9,10,11].

Such an endeavor can be pursued in models exhibiting up–down symmetry [12,13,14], as well as in models featuring absorbing states within the universality class of direct percolation [8,15,16,17,18,19]. Additionally, these studies extend to tricritical points (TCP), encompassing two-dimensional and three-dimensional short-range systems, as well as systems under mean-field approximation [7,20,21,22,23], each with their distinctive characteristics and intricacies.

Utilizing time-dependent simulations to investigate these techniques presupposes the capability to compress temporal evolution while contending with inherently large systems, a challenge frequently encountered in equilibrium simulations. Moreover, equilibrium simulations grapple with the issue of critical slowing down. Nowadays, simulations of two-dimensional systems typically encompass linear dimensions denoted by *L*, conservatively ranging in the order of hundreds or, with more generous allocations, extending into the thousands. These simulations leverage parallelization techniques and GPU applications to effectively handle computational demands.

This suggests that exploring non-conventional methods is intriguing. Upon delving into the literature, the concept of random matrices reveals fascinating aspects. Originating from the intricate description of the distribution of energy levels in heavy nuclei proposed by Wigner [24,25,26,27], its connection with the thermodynamics of Coulomb gases primarily stems from the works of Dyson [28,29,30].

Alternatively, utilizing individual time series to construct appropriate matrices, known as Wishart matrices [31,32], offers a particularly intriguing avenue for delving into the statistical mechanics of spin systems and their idiosyncrasies. This approach is underscored by recent contributions [33,34,35,36], as these matrices encapsulate the time-correlations of specific random variables described by stochastic processes, which can be modeled by Langevin equations or simulated via Monte Carlo (MC) Markov chains. It is noteworthy to mention that while the integrability aspects of lattice spin systems have been explored using random matrices, the emphasis has traditionally been on direct examination of the Hamiltonians of such systems [37], rather than on correlation matrices as in the previously mentioned works.

Pioneering contributions in the late 1980s and early 1990s from Cicuta and Molinari (see, for example, [38]) explored the relationship between the critical properties of statistical mechanics models in equilibrium and the spectral density of random matrices that culminate in a large and prolific carrier [39].

Cicuta and Molinari investigated the emergence of multicut solutions and the behavior of eigenvalue densities near critical points. This line of inquiry has been further explored by other authors, such as Eynard [40], who studied the eigenvalue distribution of a random matrix at transitions where a new connected component of the eigenvalue density support appears away from other connected components.

It is also known that the largest eigenvalue of a complex Gaussian sample covariance matrix, along with others from the Gaussian family, exhibits sharp phase transitions, which have been studied in the literature (see, for example, [41,42]). Following the initial findings by Costin and Lebowitz [43] for Gaussian matrices, which demonstrated Gaussian fluctuations in the number of eigenvalues of random matrices in windows scaled with the square root of the window size, some authors have explored the existence of a transition in the variance of the number of eigenvalues at the edge of the semi-circle law [44].

The connection between random matrices and thermodynamics is fundamentally grounded in the Eigenstate Thermalization Hypothesis (ETH), which elucidates the emergence of thermodynamic equilibrium in isolated quantum many-body systems by positing a specific structure for the matrix elements of observables in the energy eigenbasis [45].

In the pursuit of demonstrating computer methods in random matrices capable of identifying phase transitions in spin systems, some authors have delved into particularly intriguing classes of random matrices, such as correlation matrices within Wishart-like ensembles [33,34,35]. Differing from other approaches in the literature, the method advocated by one of the authors of this current work in [35] diverges significantly by eschewing the use of matrices whose dimensions directly correspond to the number of lattice sites. While approaches employing matrices scaled according to the number of lattice sites are theoretically sound and intriguing, they can prove computationally prohibitive due to the sheer scale of the involved matrices. Instead, the proposed method operates effectively with dimensions in the order of a few hundred, derived from various short-term evolution samples of the magnetization system. This strategy, as supported by our previous research [35,36], demonstrates both computational efficiency and theoretical robustness.

In this study, we aim to extend the applicability of random matrix methodology by investigating models that exhibit tricritical points, with a particular emphasis on the transition from critical points to the tricritical one. Through careful analysis of the spectra derived from Wishart-like matrices, which capture the correlations among diverse magnetization time series, we showcase the effectiveness of our approach in elucidating the phase diagram of the Blume–Capel (BC) model in both two and three dimensions. Throughout this paper, we present our findings, highlighting how this methodology adeptly reveals the intricate characteristics of complex systems like the BC model.

Furthermore, we demonstrate that this method, despite its efficiency and suitability for short times, operates effectively for small systems, in contrast to the standard MC method.

In the next section, we introduce the Wishart-like method alongside a concise overview of the established properties of the BC model. It is crucial to emphasize that our primary focus lies not on the intricacies of the model itself. Rather, our goal was to select the simplest model featuring a tricritical point in both two and three dimensions to validate our study.

In the following section (Section 3), we unveil our principal findings, demonstrating the efficacy of our method in accurately pinpointing the critical points of both the two-dimensional and three-dimensional versions of the model. Additionally, we illustrate how our approach adeptly captures the crossover effects in a spectral manner. Furthermore, a novel aspect explored in this study, not previously investigated in our prior works, is the successful application of extreme value statistics in establishing the criticality of such systems.

Lastly, we demonstrate the effectiveness of our method, even for very small systems. We illustrate that although there may be a computational cost associated with diagonalizing matrices, this cost is offset by the advantages gained from shorter computation times and smaller system sizes. The paper concludes with a summary of key findings and conclusions.

## 2. Random Matrices, Critical, and Tricritical Points in the
Blume–Capel Model

Tricritical points play a vital role in contemporary research within condensed matter theory and statistical mechanics. The pioneering discovery of the first tricritical point in He-3 and He-4 mixtures by Griffiths in 1970 marked a significant milestone [46]. Subsequently, Griffiths, along with Blume and Emery, introduced the BEG (Blume–Emery–Griffiths) model in 1971, which provided a framework to replicate the thermodynamic behavior observed in these mixtures [47]. This model, based on a spin-1 Ising model, has since become a cornerstone in the study of tricritical phenomena.

In its broader scope, the Hamiltonian can be expressed as:$$H=-K\sum_{\langle ij\rangle }s_{i}^{2}s_{j}^{2}-H_{3}\sum_{\langle ij\rangle }s_{i}s_{j}\left(s_{i}+s_{j}\right)-J\sum_{\langle ij\rangle }s_{i}s_{j}+D\sum_{i=1}^{N}s_{i}^{2}-H\sum_{i=1}^{N}s_{i}.$$

The initial term delineates the crystalline interaction among the spins, whereas the subsequent term denotes the multispin interaction among them. Finally, the third, fourth, and last terms represent the Ising interaction between spin pairs, anisotropic interaction, and the interaction of spins with an external magnetic field, respectively.

Indeed, an even more straightforward variation of this model, commonly referred to as the Blume–Capel (BC) model [48,49], is delineated by the Hamiltonian:(1)$$H=-J\sum_{\langle i,j\rangle }s_{i}s_{j}+D\sum_{i}s_{i}^{2}-H\sum_{i}s_{i},$$
where each spins can hold the values $s_{i}\in \left\{-1,0,+1\right\}$; this would be enough to present the existence of a tricritical point in both two and three dimensions, separating a critical line of the first-order line.

The first term models the local interaction between the spins, with J>0 representing the interaction strength and 〈i,j〉 indicating that the interaction occurs between nearest neighbor pairs of sites *i* and *j*. The parameter *D* is called the anisotropy field and is responsible for zero-field splitting, resulting in an increase in energy for $s_{i}=\pm 1$ states, even in the absence of an external magnetic field. Finally, the third term models the interaction of the system with an external magnetic field of intensity *H*, which we will assume is not present.

In this scenario, the model delineates a critical line (CL) culminating in a tricritical point (TCP). Subsequently, it exhibits a first-order line (FOL), as illustrated in Figure 1 for both the two-dimensional and three-dimensional versions of the BC model in the absence of an external magnetic field.

This figure is utilized only for pedagogical reasons in this work since all points used in this current work uses the points estimated in this line that were very didactically obtained by Butera and Pernici [50]. Certainly, a significant number of estimates shown in this reference were re-obtained by the authors since there had been an evolution of estimates with very different methods over the years that included many good works (see, for example [51,52,53,54]). Thus, the question is whether or not we can detect the critical line of the BC model for different values of *D* using Wishart-random matrices spectra and how the method responds to the crossover between CL and FOL intermediated by the TCP.

Therefore, leveraging a framework established in prior research, this study demonstrates our capability to localize critical points within two- and three-dimensional BC models, while also investigating the existing crossover phenomena. Subsequently, the following subsections provide a concise overview of the random matrix methodology employed in our approach, along with the quantities slated for estimation within this method.

### 2.1. Random Matrices and Phase Transitions: General Comments

An ensemble of symmetric n×n matrices with a probability density that remains invariant under orthogonal transformations possesses a joint distribution given by:$$\mathrm{Pr}\left(H\right)=\mathrm{Pr}\left(H_{11},H_{12}\ldots ,H_{nn}\right)=\mathrm{Pr}\left({\left\{H_{ij}\right\}}_{i\leq j}\right)=\frac{1}{Z_{n}}e^{-Tr(V(H))},$$
such that $Z_{n}=\int \ldots \int e^{-Tr(V(H))}dH_{11}\ldots dH_{nn}$. In this case, the distribution of eigenvlaues is written as:$$\mathrm{Pr}\left(\lambda_{1},\ldots ,\lambda_{n}\right)=\frac{1}{Z_{N}}e^{-H(\lambda_{1},\ldots ,\lambda_{n})},$$
where now $Z_{n}=\int \ldots \int d\lambda_{1}\ldots d\lambda_{n}e^{-H(\lambda_{1},\ldots ,\lambda_{n})}$, where:$$H\left(\lambda_{1},\ldots ,\lambda_{n}\right)=-\sum_{i<j}\ln\left|\lambda_{i}-\lambda_{j}\right|+\sum_{i=1}^{n}V\left(\lambda_{i}\right),$$
by introducing a term reflecting the repulsion among the eigenvalues, alongside a second term dependent on the potential V(λ), which signifies the interaction of the particles with an external field.

If we consider symmetric ($H_{ij}=H_{ji}$) and well-behaved entries, i.e., distributed according to a probability density function, f(H), such that
$$\int_{-\infty }^{\infty }dH_{ij}f\left(H_{ij}\right)H_{ij}^{k}<\infty ,$$
for k=1,2, of a matrix, *H*, with dimension n×n, and independent entries, and therefore with joint distribution given by:$$\begin{array}{ccc} \mathrm{Pr}(H_{11},H_{12}\ldots ,H_{nn}) & = & \mathrm{Pr}\left({\left\{H_{ij}\right\}}_{i\leq j}\right)\\ \; & \; & \\ \; & = & \prod_{i\leq j}f\left(H_{ij}\right). \end{array}$$

In the particular case that $f\left(h_{ij}\right)=\frac{e^{-h_{ij}^{2}/2}}{\sqrt{2\pi }}$, one has the Boltzmann weight:$$P\left(\lambda_{1},\ldots ,\lambda_{N}\right)=Z_{n}^{-1}\exp\left[-\frac{1}{2}\sum_{i=1}^{n}\lambda_{i}^{2}+\sum_{i<j}\ln\left|\lambda_{i}-\lambda_{j}\right|\right],$$
where $Z_{n}=\int_{0}^{\infty }\ldots \int_{0}^{\infty }d\lambda_{1}\ldots d\lambda_{n}\exp\left[-H\left(\lambda_{1}\ldots \lambda_{n}\right)\right]$, where in this case $V\left(\lambda \right)=\frac{1}{2}\lambda^{2}$, which corresponds to $V\left(H\right)=\frac{1}{2}tr\;H^{2}$. The eigenvalue density
$$\rho \left(\lambda \right)=\int_{-\infty }^{\infty }\ldots \int_{-\infty }^{\infty }P\left(\lambda ,\lambda_{2},\lambda_{3},\ldots ,\lambda_{n}\right)d\lambda_{2}\ldots d\lambda_{n}$$
is universally described by the semi-circle law [27]:(2)$$\rho \left(\lambda \right)=\left\{\begin{array}{c} \frac{1}{\pi }\sqrt{2n-\lambda^{2}}\;\;\;\mathrm{if}\;\;\;\;\lambda^{2}<2n\\ 0\;\;\;\;\mathrm{if}\;\;\;\;\lambda^{2}\geq 2n \end{array}\right..$$

It’s pivotal to acknowledge that representing V(λ) as a quadratic form lays the groundwork for a resilient type of central limit theory that is applicable to random matrices. Simply stating that the matrix entries are symmetric, independent, identically distributed, and have well-defined first and second moments, as observed by Sinai and Soshnikov [55], is sufficient to demonstrate universal behavior in this case. However, deviations from the density of states given by Equation (Equation 2) may depend on the potential V(λ), which is contingent upon the matrix entries. Cicuta and Molinari investigated the emergence of multicut solutions and the behavior of eigenvalue densities near critical points [39]. In the simplest one-matrix model of spectral density (see [56]), the spectral density is expressed in a more general manner:$$\rho \left(\lambda \right)=\left\{\begin{array}{c} \frac{1}{\pi }f\left(\lambda \right)\sqrt{2n-\lambda^{2}}\;\;\;\mathrm{if}\;\;\;\;\lambda^{2}<2n\\ 0\;\;\;\;\mathrm{if}\;\;\;\;\lambda^{2}\geq 2n, \end{array}\right.$$
where f(λ) is determined by the potential V(λ). For example, considering a general quartic potential $V\left(\lambda \right)=a_{1}\lambda +a_{2}\lambda^{2}+a_{4}\lambda^{4}$. When $a_{1}=0$, for $a_{2}\geq -2$ there is a 1-cut solution. At $a_{2}=-2$, the density has a zero in the middle of its support. On the other hand, for $a_{2}\leq -2$ one must consider a two-cut solution, given by:$$\rho \left(\lambda \right)\approx \left|\lambda \right|\sqrt{\lambda_{+}^{2}-\lambda^{2}}\sqrt{\lambda^{2}-\lambda_{-}^{2}},$$
with $\lambda_{+}$ and $\lambda_{-}$ being, respectively, the upper and lower bound values.

The same authors go further by demonstrating that there exists a diagram with $a_{1}$ plotted against $a_{2}$, which exhibits three distinct phases. The first phase entails a one-cut solution, the second reveals a two-cut solution, and finally, the third phase depicts a coexistence between these two possibilities.

However, for a special class of random matrices—correlation random matrices, upon which our method is based—there are significant idiosyncrasies. In the next section we will explore these idiosyncrasies by presenting our method, which is based on how the gap of eigenvalues in the eigenvalue density depends on the temperature of the system and how it governs the criticality of spin systems

More precisely, our assumption is founded on the notion that the critical behavior of the spin system under investigation is reflected in the critical behavior of a Coulomb gas. This inference stems from the joint distribution of eigenvalues serving as the Boltzmann weight of the Hamiltonian governing this Coulomb gas, which inherently relies on the temperature of the spin system. Essentially, the existing correlations within the random matrices will impact the potential of the Coulomb gas and, consequently, the moments of the eigenvalues.

### 2.2. Wishart-like Matrices and Spin Systems

In our analysis, we introduce the magnetization matrix element $m_{tj}$, which denotes the magnetization of the *j*th time series at the *t*th Monte Carlo (MC) step in a system comprising $N=L^{d}$ spins. For simplicity, we set d=2, as it represents the minimal dimension for the manifestation of phase transitions in short-range interaction systems. In this context, *t* ranges from 1 to $N_{MC}$ and *j* ranges from 1 to $N_{sample}$, thereby constructing the magnetization matrix, *M*, with dimensions $N_{MC}\times N_{sample}$.

To delve into the spectral properties, an interesting approach is to shift our focus away from *M* and instead examine the square matrix of size $N_{sample}\times N_{sample}$:$$G=\frac{1}{N_{MC}}M^{T}M\;,$$
where each element $G_{ij}$ of *G* is defined as $G_{ij}=\frac{1}{N_{MC}}\sum_{t=1}^{N_{MC}}m_{ti}m_{tj}$, referred to as the Wishart matrix [31]. To simplify computations, it is advantageous to transform the components of matrix *M* using the transformed matrix $M^{*}$, whose elements are expressed in terms of standard variables as follows:$$m_{tj}^{*}=\frac{m_{tj}-{\langle m_{j}\rangle }_{t}}{\sqrt{{\langle m_{j}^{2}\rangle }_{t}-{\langle m_{j}\rangle }_{t}^{2}}},$$
where ${\langle m_{j}^{k}\rangle }_{t}=\frac{1}{N_{MC}}\sum_{i=1}^{N_{MC}}m_{ij}^{k}\;$. This transformation facilitates subsequent analysis and calculations.

Thereby:(3)$$G_{ij}^{*}=\frac{\langle m_{i}m_{j}\rangle -\langle m_{i}\rangle \langle m_{j}\rangle }{\sigma_{i}\sigma_{j}},$$
where ${\langle m_{i}m_{j}\rangle }_{t}=\frac{1}{N_{MC}}\sum_{t=1}^{N_{MC}}m_{ti}m_{tj}$ and $\sigma_{i}=\sqrt{\langle m_{i}^{2}\rangle -{\langle m_{i}\rangle }^{2}}$. Here it is crucial to expound upon a pivotal calculation that elucidates the application of these matrices in greater detail. We consider two distinct time evolution samples of the magnetization per spin, denoted as $m_{ti}$ and $m_{tj}$, where $t=1,\ldots ,N_{MC}$. In this context:$$m_{tj}=\frac{1}{N}\sum_{k=1}^{N}\sigma_{t,j,k},$$
where $\sigma_{t,j,k}$ denotes the value of the *k*-th spin in the *j*-th evolution or run at time *t*.

We can establish the correlation between these two time series using the following definition:$$\begin{array}{ccc} {\langle m_{i}m_{j}\rangle }_{t} & = & \frac{1}{N_{MC}}\sum_{t=1}^{N_{MC}}m_{ti}m_{tj}\\ \; & \; & \\ \; & = & \frac{1}{N^{2}N_{MC}}\sum_{t=1}^{N_{MC}}\left(\sum_{k=1}^{N}\sigma_{t,i,k}\right)\left(\sum_{l=1}^{N}\sigma_{t,j,l}\right)\\ \; & \; & \\ \; & = & \frac{1}{N^{2}N_{MC}}\sum_{t=1}^{N_{MC}}\left(\sum_{k=1}^{N}\sigma_{t,i,k}\sigma_{t,j,k}+\sum_{k\neq l=1}^{N}\sigma_{t,i,k}\sigma_{t,j,l}\right)\\ \; & \; & \\ \; & = & \frac{1}{N^{2}}\sum_{k=1}^{N}\left[\left(\frac{1}{N_{MC}}\sum_{t=1}^{N_{MC}}\sigma_{t,i,k}\sigma_{t,j,k}\right)+\sum_{k\neq l=1}^{N}\left(\frac{1}{N_{MC}}\sum_{t=1}^{N_{MC}}\sigma_{t,i,k}\sigma_{t,j,l}\right)\right]\\ \; & \; & \\ \; & = & \frac{1}{N^{2}}\sum_{k=1}^{N}{\langle \sigma_{i,k}\sigma_{j,k}\rangle }_{t}+\frac{1}{N^{2}}\sum_{k\neq l=1}^{N}{\langle \sigma_{i,k}\sigma_{j,l}\rangle }_{t}. \end{array}$$

Given that $\sum_{k=1}^{N}{\langle \sigma_{i,k}\sigma_{j,k}\rangle }_{t}=O\left(N\right)$ and $\sum_{k\neq l=1}^{N}{\langle \sigma_{i,k}\sigma_{j,l}\rangle }_{t}=O\left(N^{2}\right)$, it follows that the thermodynamic limit is (N→∞):$${\langle m_{i}m_{i}\rangle }_{t}\approx {\langle \frac{1}{N^{2}}\sum_{k\neq l=1}^{N}\sigma_{i,k}\sigma_{j,l}\rangle }_{t}=\frac{1}{N^{2}}{\langle \sigma_{i}⊗\sigma_{j}\rangle }_{t}.$$

When $T>T_{C}$, ${\langle m_{i}\rangle }_{t}\approx 0$. This leads to: ${\langle m_{i}m_{j}\rangle }_{t}-{\langle m_{i}\rangle }_{t}{\langle m_{j}\rangle }_{t}\approx {\langle m_{i}m_{j}\rangle }_{t}=\frac{1}{N^{2}}{\langle \sigma_{i}⊗\sigma_{j}\rangle }_{t}$, and we can express the correlation coefficient (our matrix element of *G*) as:$$\begin{array}{ccc} G_{ij}^{*} & \approx & \frac{{\langle m_{i}m_{j}\rangle }_{t}}{\sqrt{{\langle m_{i}^{2}\rangle }_{t}}\sqrt{{\langle m_{j}^{2}\rangle }_{t}}}\\ \; & \; & \\ \; & = & \frac{{\langle m_{i}m_{j}\rangle }_{t}}{{\langle m_{i}^{2}\rangle }_{t}}\\ \; & \; & \\ \; & = & \frac{{\langle \sigma_{i}⊗\sigma_{j}\rangle }_{t}}{{\langle \sigma_{i}⊗\sigma_{i}\rangle }_{t}}, \end{array}$$
where $\sigma_{i}\equiv \left(\sigma_{i,1},\ldots ,\sigma_{i,N}\right)$ and $\sigma_{j}\equiv \left(\sigma_{j,1},\ldots ,\sigma_{j,N}\right)$. Thus, $g_{ij}$ for $T>T_{C}$ is determined by:$$G_{ij}^{*}=\frac{{\langle \sigma_{i}⊗\sigma_{j}\rangle }_{t}}{{\langle \sigma_{i}⊗\sigma_{i}\rangle }_{t}}.$$

This metric assesses the relationship between the temporal averages of spatial correlations within both inter- and intra-time series. By analyzing both spatial and temporal dimensions, it provides a compelling approach to delve into spin systems.

Thinking in the general case, when the variables $m_{ij}^{*}$ are uncorrelated random variables, momentarily forgetting the context of these variables represents the magnetization of spin system; the eigenvalue density, ρ(λ), of the matrix $G^{*}=\frac{1}{N_{MC}}M^{*T}M^{*}$ conforms to the well-known Marchenko–Pastur (MP) distribution [57]. For our specific case, we express this distribution as:(4)$$\rho \left(\lambda \right)=\left\{\begin{array}{c} \frac{N_{MC}}{2\pi N_{sample}}\frac{\sqrt{\left(\lambda -\lambda_{-}\right)\left(\lambda_{+}-\lambda \right)}}{\lambda }\;\mathrm{if}\;\lambda_{-}\leq \lambda \leq \lambda_{+}\\ 0\;\;\;\;\mathrm{otherwise} \end{array}\right.,$$
where $\lambda_{\pm }=1+\frac{N_{sample}}{N_{MC}}\pm 2\sqrt{\frac{N_{sample}}{N_{MC}}}.$

Here it is important to mention that this density is obtained by integrating the joint distribution of eigenvalues, i.e., $\rho \left(\lambda \right)=\int_{-\infty }^{\infty }\ldots \int_{-\infty }^{\infty }d\lambda_{2}\ldots \lambda_{N_{sample}}P\left(\lambda ,\lambda_{2}\ldots ,\lambda_{N_{sample}}\right)$, which, in this case, is given by:$$\begin{array}{ccc} P(\lambda_{1},\ldots ,\lambda_{N_{sample}}) & = & C_{N_{sample}}\exp\left(-\frac{N_{MC}}{2}\sum_{i=1}^{N_{sample}}\lambda_{i}+\frac{N_{MC}-N_{sample}-1}{2}\sum_{i=1}^{N_{sample}}\ln\lambda_{i}\right.\\ \; & \; & \\ \; & \; & \left.+\sum_{i<j}\ln\left|\lambda_{i}-\lambda_{j}\right|\right) \end{array}$$
which results in a potential:$$V\left(\lambda \right)=\frac{N_{MC}}{2}\lambda -\frac{N_{MC}-N_{sample}-1}{2}\ln\lambda ,$$
corresponding to uncorrelated (Wishart) matrices. However, for different temperatures, this potential must vary, and we must consider V(λ|T), such that only for high temperatures V(λ|T)≈V(λ).

Undoubtedly, we expect that for $T\gg T_{c}$, the density of eigenvalues $\rho^{\exp}\left(\lambda \right)$ obtained from computational simulation approaches ρ(λ) in Equation (Equation 4), but our method may not necessarily fit such a distribution perfectly due to residual autocorrelation. The interesting question is what happens when $T\approx T_{C}$. Moreover, we will utilize the density $\rho^{\exp}\left(\lambda \right)$, obtained from computer simulations, to determine the critical parameters of spin models.

The moments of $\rho^{\exp}\left(\lambda \right)$ are calculated as:(5)$$\langle \lambda^{k}\rangle =\frac{\sum_{i=1}^{N_{\mathrm{bin}}}\lambda_{i}^{k}\rho^{\exp}\left(\lambda_{i}\right)}{\sum_{i=1}^{N_{\mathrm{bin}}}\rho^{\exp}\left(\lambda_{i}\right)},$$
where $N_{\mathrm{bin}}$ is the number of bins of the histogram of $\rho^{\exp}\left(\lambda \right)$. Thus, for $T\gg T_{c}$ we also expect $\bar{\lambda^{k}}$ to approach:Eλk=∫−∞∞λkρ(λ)dλ=NMC2πNsample∫λ−λ+λk−1(λ−λ−)(λ+−λ)dλ=∑j=0k−1NsampleNMCjj+1kjk−1j.

Explicitly, E[λ]=1 and E[λ2]=∑j=01NsampleNMCjj+12j1j=1+NsampleNMC. However, beyond these limits the behavior of $\langle \lambda^{k}\rangle$ can provide thermodynamic information about spin models, as suggested by our previous works [35,36]. In those works, we observed that monitoring $\langle \lambda \rangle$ and $\langle {\left(\Delta \lambda \right)}^{2}\rangle =\langle {\left(\lambda -\langle \lambda \rangle \right)}^{2}\rangle$ as a function of $\frac{T}{T_{C}}$ indicates a minimum of $\langle \lambda \rangle$, and an inflection point for $\langle {\left(\lambda -\langle \lambda \rangle \right)}^{2}\rangle$ (or divergence of its derivative) occurs at $T=T_{C}$.

Here, we will demonstrate that this method works effectively for the BC model in both two and three dimensions, particularly in identifying critical points and examining its response to the crossover phenomena between CL and FOL.

## 3. Results

We will now present our main results. In the first subsection, we showcase the outcomes of our spectral method concerning the critical points of the BC model in both two and three dimensions. Following this, in the second subsection, we extend our investigation to demonstrate the crossover effects in the model, as captured by the density of maximal eigenvalues of Wishart matrices.

For our analysis, we construct $N_{run}=1000$ distinct matrices $G^{*}$ of size $N_{sample}\times N_{sample}$ for each fixed temperature. Each matrix is derived from $N_{sample}=100$ magnetization time series, each comprising $N_{MC}=300$ Monte Carlo steps. These time series are obtained via MC simulations employing heatbath single spin flip dynamics for the BC model, resulting in a total of $10^{5}$ eigenvalues used to construct the histogram for each temperature. All eigenvalues are categorized into $N_{b}=100$ bins. In the two-dimensional systems, we utilize a linear dimension of L=100, while in three dimensions we employ L=22.

An essential aspect for the accurate numerical application of the method involves utilizing the histogram to compute the eigenvalue moments through numerical experiments, as per Equation Equation 5, and directly calculating the numerical moments. It is crucial to emphasize this point for readers intending to apply the method, as we have confirmed that computing the averages directly does not yield the expected results presented here.

### 3.1. Critical Points

We begin our results by displaying the histogram of eigenvalues. We choose D=1 for both the 2D and 3D BC models to illustrate the density of eigenvalues obtained through the diagonalization of matrices $G^{*}$. Figure 2 presents histograms for various temperatures. An evolution of the gap between the two eigenvalue bulks can be observed. Similar behavior is noted for the Ising model on two-dimensional lattices under mean-field approximations [35,36].

We can discern analogous behavior in the three-dimensional BC model, as illustrated in Figure 3.

It is crucial to highlight the distinctive trend of the eigenvalue gap narrowing around the critical temperature, along with the correspondence to the MP law for $T>T_{C}$. While a perfect agreement is not expected as *T* approaches infinity due to the correlation matrix’s construction, incorporating total magnetization and time series with inherent autocorrelation, it is important to note that this does not diminish the method’s validity in any manner.

However, it is necessary to utilize this density of states to effectively determine the critical parameter. This can be achieved by computing the moments of the density of states, specifically $\langle \lambda \rangle$ and $\langle {\left(\Delta \lambda \right)}^{2}\rangle$. In this regard, we observe the results for three different values of *D*. For the two-dimensional BC model (refer to Figure 4), we tested three values, D=0, D=1, and D=1.75, employing the corresponding $T_{C}$ values estimated in [50] as a basis. Similarly, for the 3D BC model, we utilized D=0, D=1, and D=2.2, as depicted in Figure 5.

We can observe a pronounced minimum in $\langle \lambda \rangle$ at $T=T_{C}$ in both the two-dimensional and three-dimensional versions of the BC model, which is related to the closing gap observed in Figure 2 and Figure 3. Additionally, an inflection point seems to be observed for the variance exactly at $T=T_{C}$ in both versions of the model (in two and three dimensions), demonstrating that both spectral measures—the average and variance—are effective in exploring criticality. The inset plot displays the first derivative of the variance:$$\alpha =\frac{d\langle {\left(\Delta \lambda \right)}^{2}\rangle }{dt}=T_{C}\frac{d\langle {\left(\Delta \lambda \right)}^{2}\rangle }{dT},$$
indicating that the critical temperature is associated with a pronounced minimum (a negative value of significant magnitude), where $t=\frac{T}{T_{C}}$.

To better understand such behavior, we examine the second derivative:$$\zeta =\frac{d^{2}\langle {\left(\Delta \lambda \right)}^{2}\rangle }{dt^{2}}=T_{C}^{2}\frac{d^{2}\langle {\left(\Delta \lambda \right)}^{2}\rangle }{dT^{2}},$$
and its plot is depicted in Figure 6 for both scenarios: the two-dimensional and three-dimensional BC models.

We notice that the critical temperature precisely aligns with the inflection point of the eigenvalue variance due to the condition ζ<0 for $T<T_{C}$ and ζ>0 for $T>T_{C}$. Understanding the nature of this inflection point warrants further investigation, prompting a thorough discussion. In our work, we provide an in-depth analysis of this aspect in Appendix A.

Thus, in this first subsection we observed that critical points of the BC model are well captured by this spectral methodology in both versions of the model: two and three dimensions. We used different parameters based on fluctuations of the eigenvalues and their convenient derivatives to conduct our analysis. Now, it is important to utilize this method to explore some nuances of points near the tricritical one. We will demonstrate how the method responds to the crossover effect.

### 3.2. Crossover Phenomena

We begin by simulating the average eigenvalue as a function of $T/T_{C}$. However, our focus now shifts to examining points near the tricritical point (TCP) to observe how the spectra of Wishart matrices behave when approaching this point alongside time series of magnetization simulated with (MC) simulations. By repeating our procedure, we initially investigate the issue in two dimensions to understand how the spectrum responds to the expected crossover phenomena in this model (refer to Figure 7). To accomplish this, we employed the values D=1.9, 1.92, 1.9336, 1.9421, 1.9501, and 1.96582 (TCP).

We notice that the minimum becomes less pronounced and deformed as we approach the TCP. However, it is interesting to note that, even for points near the TCP, the method indicates the critical point, albeit with reduced precision. Initially, the peak transforms into a shell, resembling a shoulder, and eventually, at the tricritical point, the minimum completely disappears.

This indicates that the average, which localizes the critical points well away from the TCP, strongly suffers the influence of this point, showing that the spectra of our correlation matrices precisely reflects what occurs with the thermodynamics of the model. Following this, we observe the dispersion of eigenvalues. We plot $\langle {\left(\Delta \lambda \right)}^{2}\rangle$ as a function of $T/T_{C}$ for the same values of *D* previously used to study $\langle \lambda \rangle$. This result is presented in Figure 8.

In contrast to the behavior observed with $\langle \lambda \rangle$, the quantity $\langle {\left(\Delta \lambda \right)}^{2}\rangle$ exhibits an inflection point very close to $T=T_{C}$, even for points near the TCP, i.e., the variance senses the crossover but is not completely extinguished as with the simple average.

However, at this precise juncture a peak occurs at the TCP, which, upon closer examination, appears to shift as *D* approaches $D_{TCP}$, i.e., we observe a migration of the maximum that will coincide at the critical temperature only exactly at TCP. Particularly intriguing is the observation that the migration of the maximum occurs with a decrease in its amplitude as *D* approaches $D_{TCP}$.

Now, we extend this investigation to the three-dimensional BC model. The behavior of the average as a function of $T/T_{C}$ for different values of *D* is illustrated in Figure 9.

Exactly as occurred in the two-dimensional version of the model, the method indicates the critical point but loses precision as it approaches the TCP. However, we observe that for the 3D version the minimum is more persistent, even at the TCP itself, since we do not observe a shoulder as obtained in the two-dimensional version.

Here, it is important to mention that in the three-dimensional version of the BC model the short-time regime presents a logarithmic correction [7,20], which should suggest such different behavior. However, again, for points far from the TCP, the minimum of $\langle \lambda \rangle$ always occurs at $T=T_{C}$, as the method exactly prescribes in its original proposal.

And what about the variance? Similar to what occurred in the two-dimensional version, the inflection point appears for all points. Their estimates in the vicinity of the TCP point differ slightly from the exact critical value. This further reinforces and suggests that we can use the inflection point of the spectral variance as a reliable indicator of critical phenomena (see Figure 10) far from the TCP, but crossover effects can generate small deviations around the TCP.

Crossover effects are observed in several works, and they play an important role in determining other quantities related to critical behavior, such as critical exponents. Here, we study their influence on the spectra of Wishart matrices built with time series of magnetization of the BC model.

Our results suggest that $\langle \lambda \rangle$ works very well for critical points outside the influence of the crossover, but it is not a good indicator of criticality near the TCP. In this case, we can make use of eigenvalue variance, which exhibits an inflection point at the critical temperature and responds reasonably well even when near the TCP, although it is also sensitive to crossover effects.

It is important to mention that MC simulations, whether in equilibrium or nonequilibrium, are generally sensitive to crossovers. For example, the dynamic exponent *z*, expected to be universal, is significantly influenced along the critical line in two dimensions [22,23], and even in the mean-field regime [21].

In Statistical Mechanics, the role of the maximum eigenvalue appears in many contexts, and an important question is whether they can also be used to quantify critical phenomena in the spectral method developed here. In other words, does the maximum eigenvalue of Wishart matrices respond to the critical behavior of the BC model? The answer is positive, and we will present the results in the next subsection.

### 3.3. Analyzing Extreme Statistics of Correlation Magnetization
Matrices

The utilization of extreme values has been extensively investigated within formal contexts to characterize phase transitions in random matrices (see, for instance, [41,42]). Nevertheless, we posit that our approach holds promise for extension, leveraging similar principles computationally and efficiently to pinpoint critical points. Hence, this paper embarks on an exploration of extreme value statistics as indicators of critical points within the BC model, employing our correlation magnetization matrices.

Accordingly, for each matrix $G^{*}$ constructed, we extract its maximum eigenvalue and compute the average across multiple runs using the following formula:$$\langle \lambda_{\max}\rangle =\frac{1}{N_{run}}\sum_{i=1}^{N_{run}}\lambda_{\max}^{\left(i\right)},$$
and we consider its behavior as a function of different temperatures for the BC model in both two and three dimensions. For such analysis, we choose D=0, 0.5, 1.0, 1.75, 1.9, and 1.92 in two dimensions and D=0, 1, 1.5, 2, 2.4, and 2.52513 for the three-dimensional version of the model. The behavior of $\langle \lambda_{\max}\rangle$ as a function of T/TC is shown in Figure 11 and Figure 12, respectively, for the cases of the two- and three-dimensional BC models.

We observe that, in both situations, the critical point is identified by a notable inflection point. Additionally, we show the first derivative in relation to $\frac{T}{T_{C}}$, simply described as $T_{C}\frac{d\langle \lambda_{\max}\rangle }{dT}$, as a function of $\frac{T}{T_{C}}$ as inset plots in these figures.

Thus, we can also observe that the averaged maximum eigenvalue responds to the criticality of the system for the different critical points studied here for the BC model, regardless of dimensionality. This adds an additional parameter to our framework to identify criticality in spin systems that can be tested in other models. In the next section, we will conclude our analysis by showing that the method that uses short times (in this current contribution and in the previous ones [35,36] we used $N_{MC}=300$ steps) also works with small systems. Up to now, we have used L=100 in two dimensions. We will demonstrate that this number can be further reduced.

### 3.4. Finite Size Scaling: Exploring Small Systems with Short Time Scales

The method’s efficiency in saving computer time through the use of short time scales presents a particularly intriguing prospect. For instance, in this current study we employed $N_{MC}=300$ steps. Therefore, to highlight the versatility of our method, we will explore another aspect: the system size. We have investigated this aspect in both two and three dimensions, demonstrating that systems can be studied effectively with even smaller sizes, yielding good estimates.

We deliberately selected only the case of D=0 without loss of generality. The average eigenvalue is plotted as a function of $T/T_{C}$ for different sizes of the two-dimensional BC system (refer to Figure 13). Initially, we explore sizes ranging from L=2 to L=16, and subsequently extend to L=20, 25, 30, 32, 64, 100, and 128.

We can observe an influence of the system size, where for small systems a minimum at the exact $T_{C}$ is found. The inset plot illustrates that for L≥32 the minimum at $T=T_{C}$ coincides. With L≥64, there is excellent agreement.

For the three-dimensional model, our investigation yields similar results. Encouragingly, we found consistent behavior, particularly noteworthy for L≥16, where a distinct trend emerges: the average eigenvalue reaches a minimum precisely at $T=T_{C}$. This observation implies the feasibility of exploring intricate phenomena within compact systems. Thus, the potential for fruitful spectral analyses in modest-scale systems becomes increasingly evident.

As we conclude this subsection, it is remarkable to note that in addition to employing short-time evolution of magnetization ($N_{MC}=300$ steps), we can also leverage small systems to achieve robust results. Surprisingly, for L≥64 in the two-dimensional BC model and L≥16 in the three-dimensional BC model (refer to Figure 14), our method accurately identifies the critical temperature of the model. This underscores the efficacy of the spectral method, highlighting its strength in pinpointing critical points.

Readers are encouraged to juxtapose our method with simpler MC simulations, considering its additional workload in terms of matrix diagonalization. However, it operates on low-dimensional matrices (with $N_{sample}=100$ here—adjustable for further optimization). In comparison with time-dependent simulations, which may require, for instance, L=256 and a substantial number of runs to adequately sample quantities (with a minimum of 2000 runs for ferromagnetic initial states and over $10^{4}$ for disordered initial states where $m_{0}\approx 0$), our method presents an intriguing alternative. It is worth noting that capturing the thermodynamics of the model with such accuracy using a computationally “cheap” spectral method is not a trivial achievement.

Equilibrium MC simulations are plagued by the issue of critical slowing down, compounded by the use of larger lattices than those employed in our approach. While a thorough comparison between this spectral method and standard MC methods warrants attention, we must emphasize the compelling observation that the thermodynamics of the systems are remarkably well-reflected by this “spectral thermodynamics”.

An avenue ripe for exploration is the investigation of long-range systems, which will undoubtedly command our focus in future applications, precisely due to the lack of requirement for large-scale systems.

## 4. Conclusions

In this study, we have extended a method originally developed in [35] to describe the spin-1 Ising model with anisotropy, known as the Blume–Capel model. This model exhibits a tricritical point in both two and three dimensions. Our method has proven effective in accurately capturing these critical points and illustrating the associated crossover phenomena.

Furthermore, we underscore the computational efficiency of our proposed method compared to similar approaches. By diagonalizing matrices of size $O(N_{sample})$, where $N_{sample}$ is set to 100 in this work, we alleviate the computational burden. This stands in contrast to other spectral methods in the literature, which necessitate diagonalizing matrices of size $O(L^{d})$, where *L* represents the linear dimension of the system and *d* its dimensionality. For instance, in a system with L=100 and d=2, this would entail diagonalizing matrices of size $10^{4}\times 10^{4}$ for a significant number of runs, which is computationally intensive.

In summary, our findings demonstrate that spectral methods provide a promising avenue for characterizing the thermodynamics of spin systems exhibiting tricritical points and crossover phenomena, regardless of the system’s dimensionality. Notably, we achieved these results using very small systems and short time series, suggesting a means to bypass both critical slowing down and the necessity for extremely large systems often observed in standard MC simulations that do not involve the diagonalization of Wishart matrices.

While our proposal does not seek to directly compete with standard MC simulations, either in equilibrium or nonequilibrium settings, our results indicate that the method merits consideration for application in these contexts due to its efficiency and sensitivity.

Lastly, we emphasize the remarkable success of the developed model in characterizing chaos [58], as well as in describing the aging effects in spin systems [59], underscoring the ongoing exploration of its full potential and the depth of understanding yet to be achieved in this research domain.

## Figures and Tables

**Figure 1 entropy-26-00395-f001:**
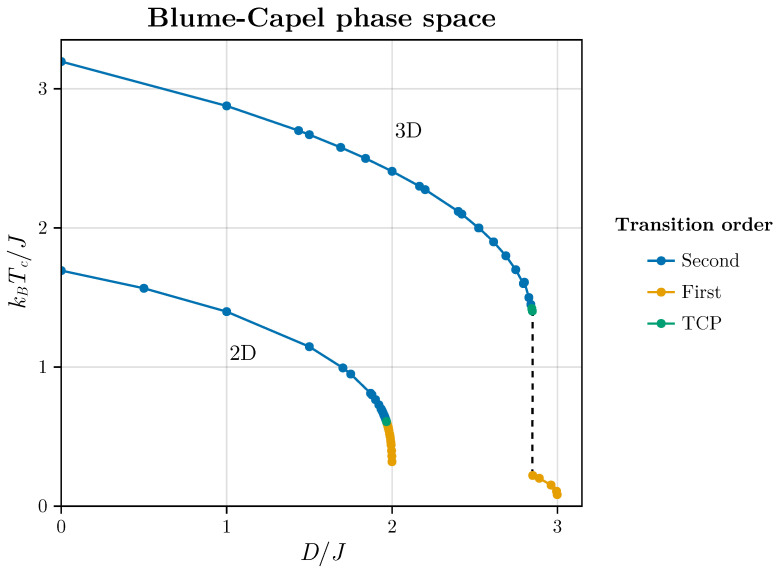
The phase diagrams for the two- and three-dimensional BC models are depicted. The points utilized in our numerical experiments are extracted from Butera and Pernici [50], serving as foundational data for the investigations conducted in this study.

**Figure 2 entropy-26-00395-f002:**
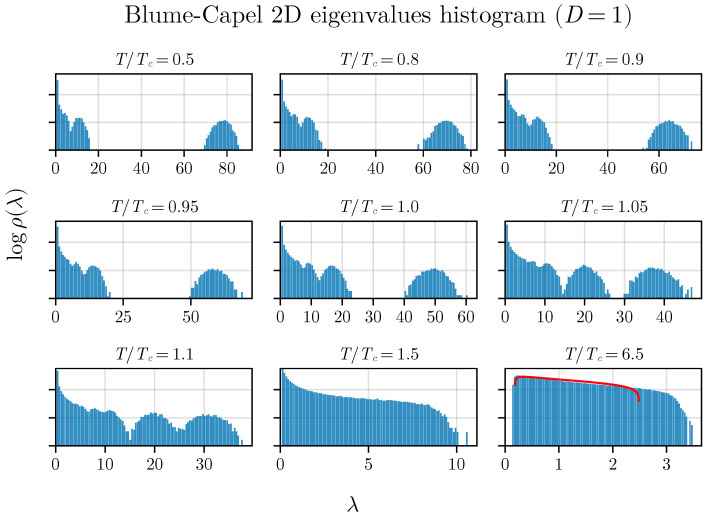
The density of states in the two-dimensional BC model with anisotropy (D=1). The gap between eigenvalues varies with the temperature of the simulated system. While the system approaches the MP law, an exact match is not achieved at high temperatures ($T>T_{C}$) due to the presence of spin–spin correlations, preventing complete correspondence.

**Figure 3 entropy-26-00395-f003:**
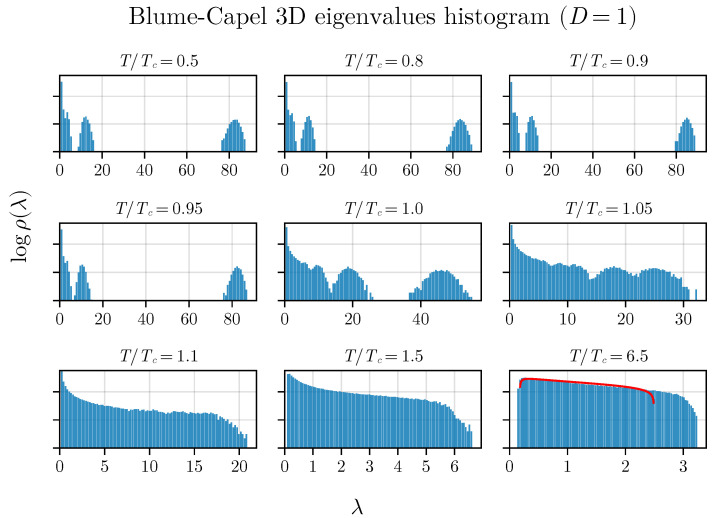
The density of states for anisotropy D=1 in the three-dimensional BC model. Similar behavior in the gap between two bulk eigenvalues is observed compared to the two-dimensional BC model (see Figure 2).

**Figure 4 entropy-26-00395-f004:**
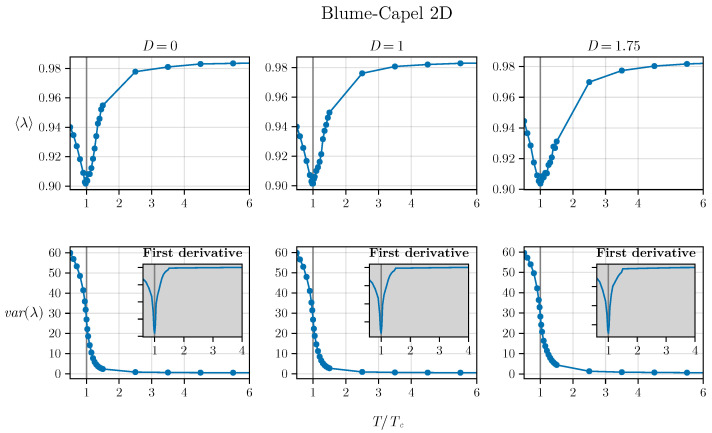
Average and variance of the two-dimensional BC model as a function of temperature are depicted. The inset plots show the derivative of the variance, indicating a divergence at $T=T_{C}$.

**Figure 5 entropy-26-00395-f005:**
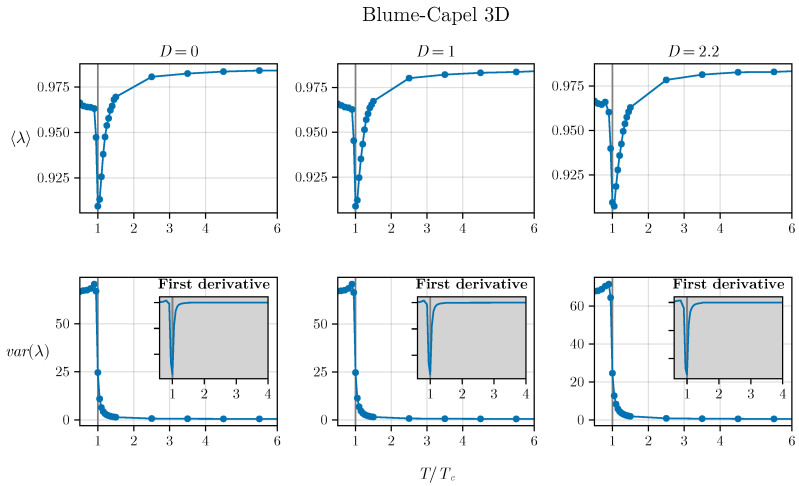
The average and variance of the three-dimensional BC model as a function of temperature illustrate a similar behavior occurring in three dimensions. The inset plots depict the derivative of the variance, highlighting its divergence at $T=T_{C}$. Interestingly, it is observed that the inflection point appears to be even more pronounced in three dimensions.

**Figure 6 entropy-26-00395-f006:**
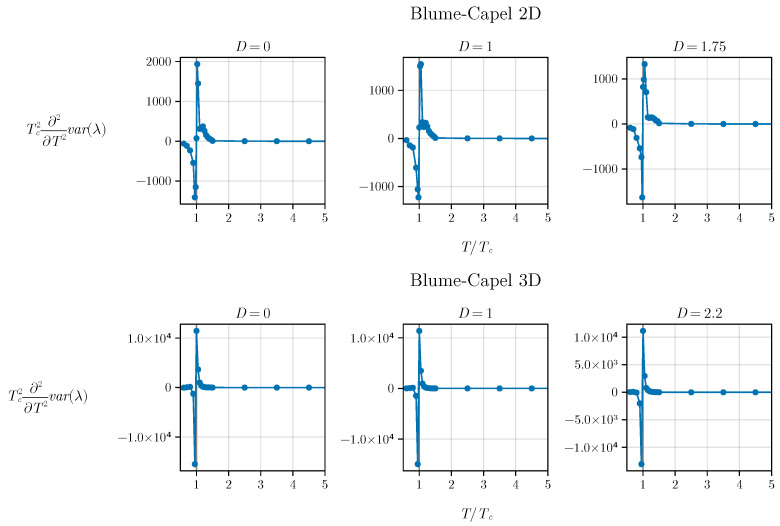
Second derivative of variance (ζ) for both the two-dimensional and three-dimensional BC models. The critical temperature precisely corresponds to the inflection point of the eigenvalue variance, indicated by ζ<0 for $T<T_{C}$ and ζ>0 for $T>T_{C}$.

**Figure 7 entropy-26-00395-f007:**
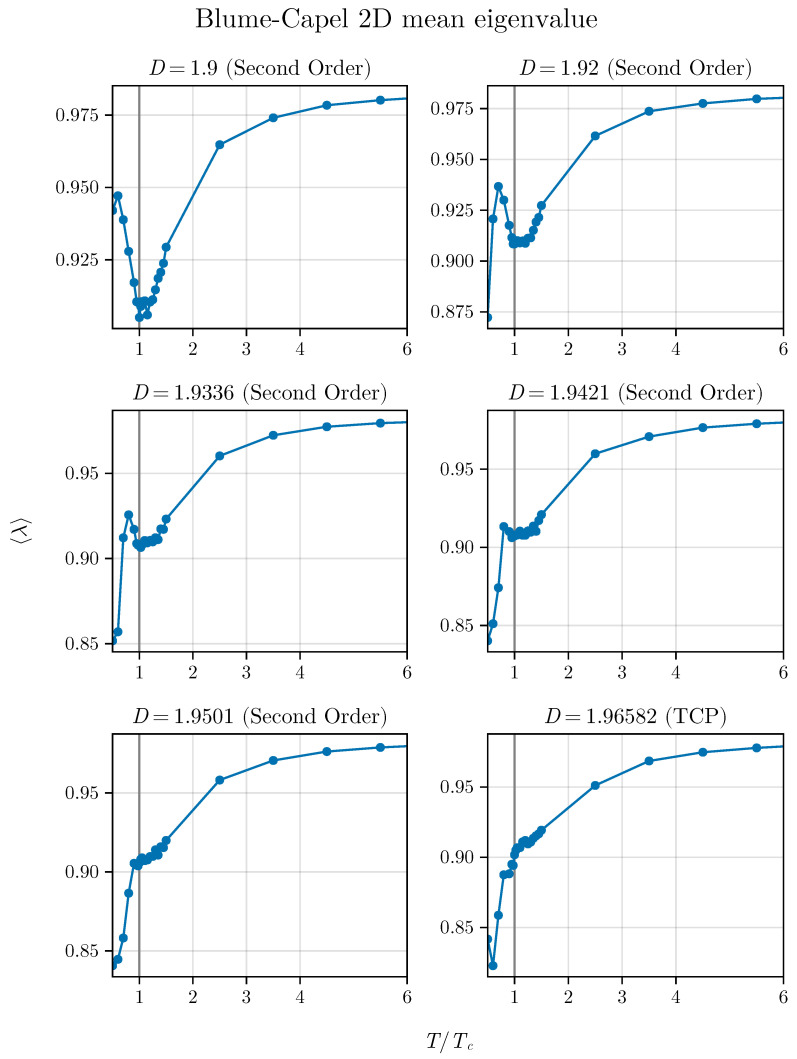
Average eigenvalue approaching the TCP in the two-dimensional BC model. We can observe that the shape of the curve is deformed as we approach the TCP on the critical line.

**Figure 8 entropy-26-00395-f008:**
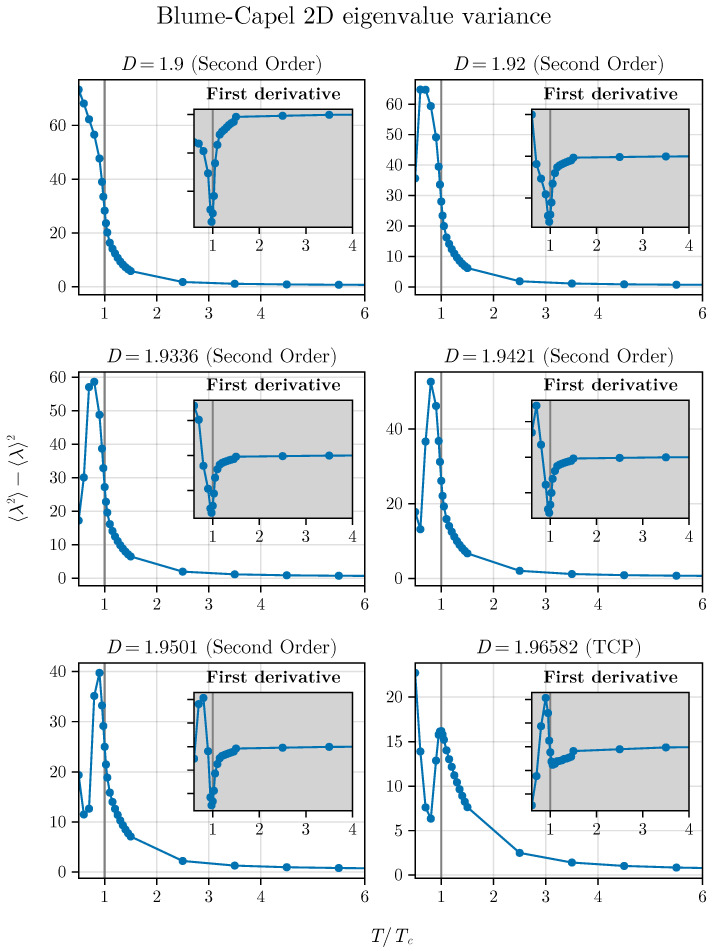
Eigenvalue variance as a function of temperature approaches the TCP in the 2D BC model. The method appears to reasonably respond even for points closer to the TCP. We can observe the inflection point up to just before the TCP, but we also notice a small deviation between the critical exact values and those determined by the method due to the crossover. At this precise TCP, there is a peak at the tricritical temperature that shifts from the previous points. Interestingly, at the TCP we do not observe the inflection point in two dimensions.

**Figure 9 entropy-26-00395-f009:**
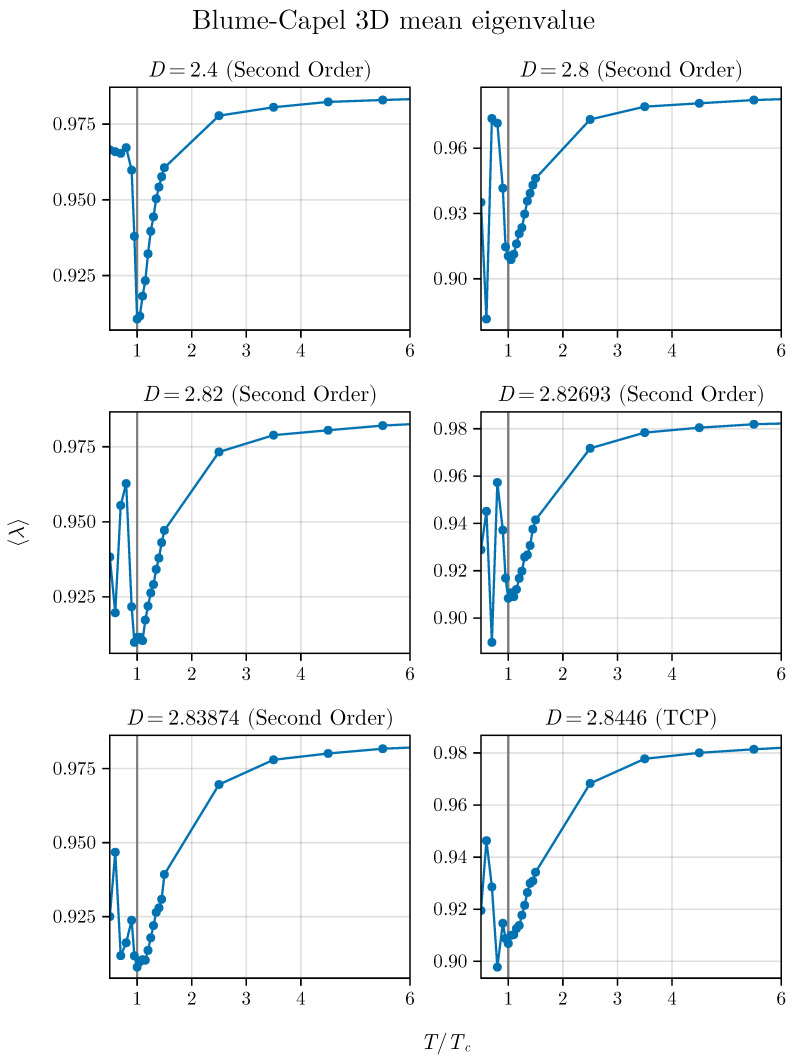
Average eigenvalue approaching the TCP in the 3D BC model, mirroring the analysis conducted for the 2D version shown in Figure 7.

**Figure 10 entropy-26-00395-f010:**
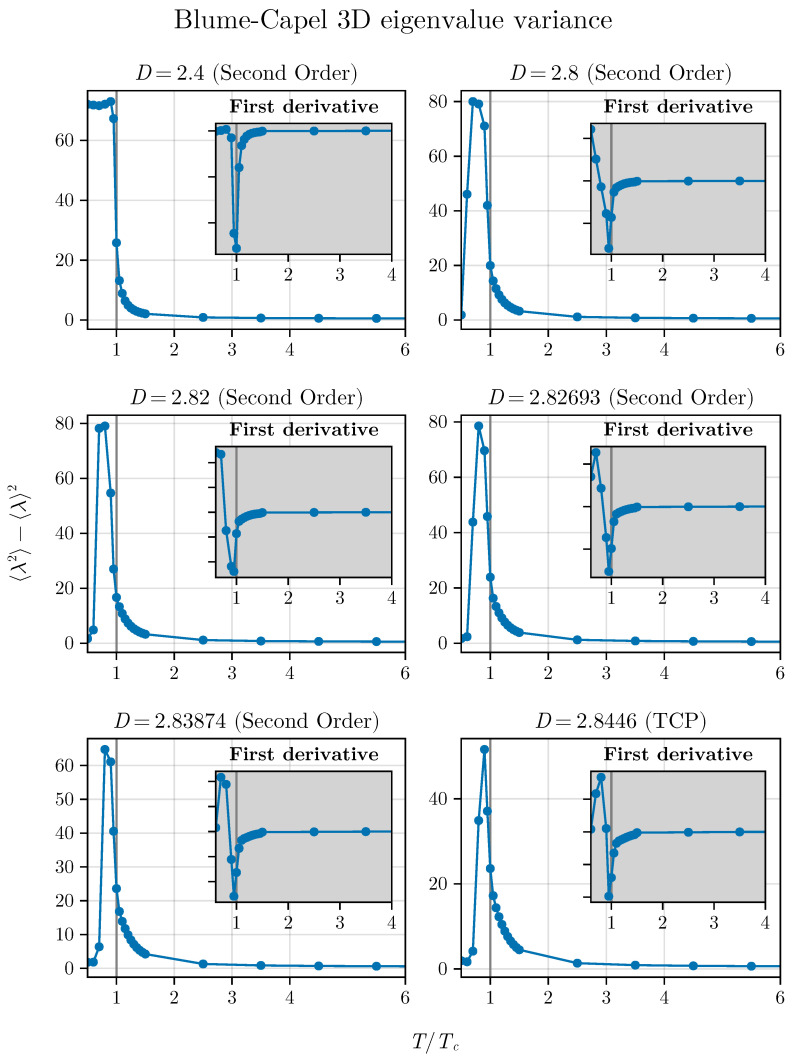
Eigenvalue variance approaching the TCP in the 3D BC model. We can observe the inflection point until the TCP, but that slightly differs from the best estimates of the critical temperatures in this vicinity of the TCP.

**Figure 11 entropy-26-00395-f011:**
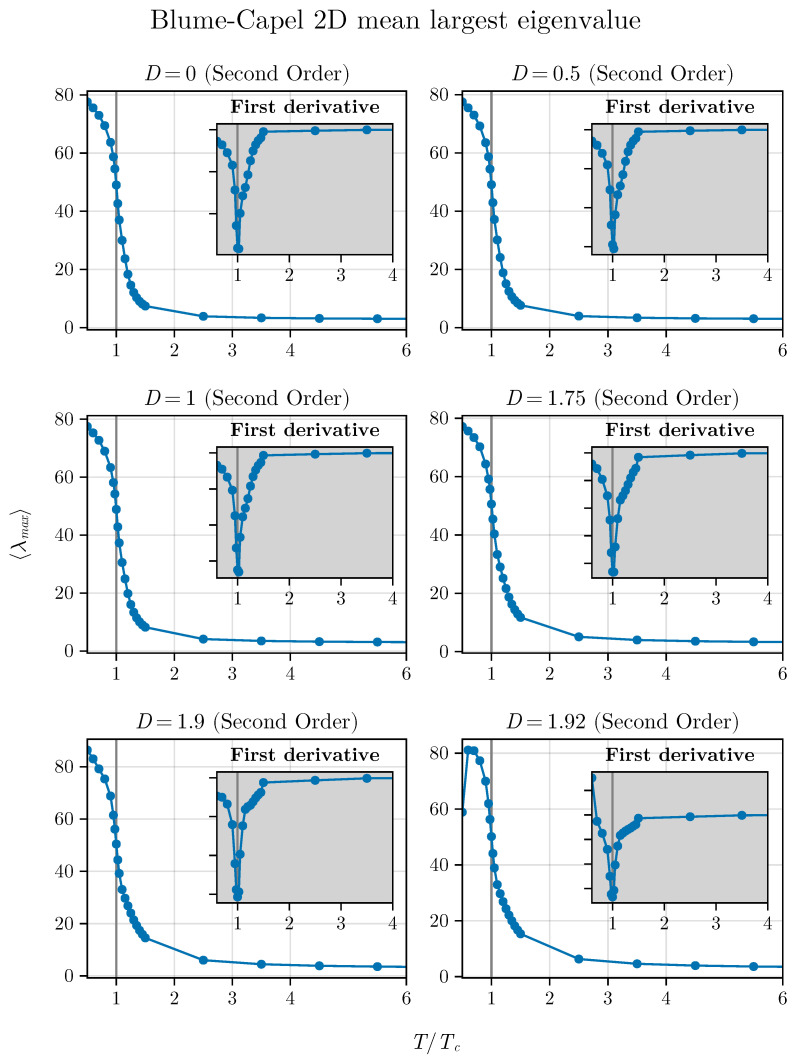
Averaged maximum eigenvalue as a function of $T/T_{C}$ for different values of *D* in the two-dimensional BC model.

**Figure 12 entropy-26-00395-f012:**
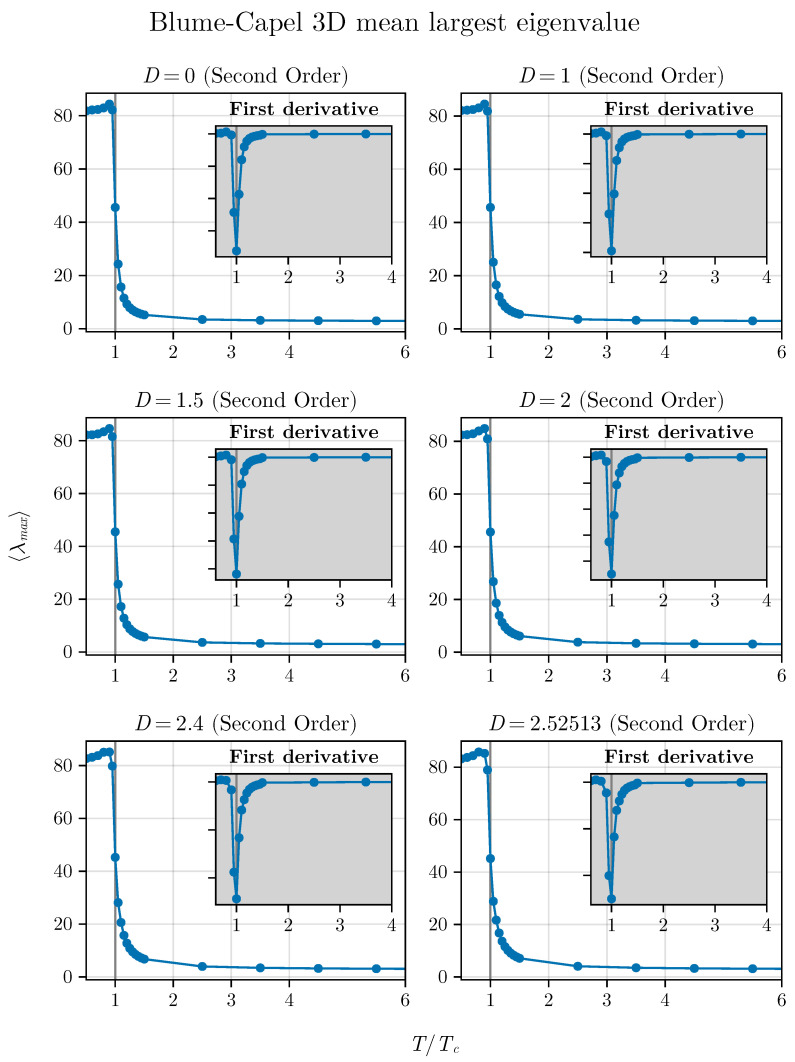
Averaged maximum eigenvalue as function of $T/T_{C}$ for different values of *D* in the three-dimensional BC model.

**Figure 13 entropy-26-00395-f013:**
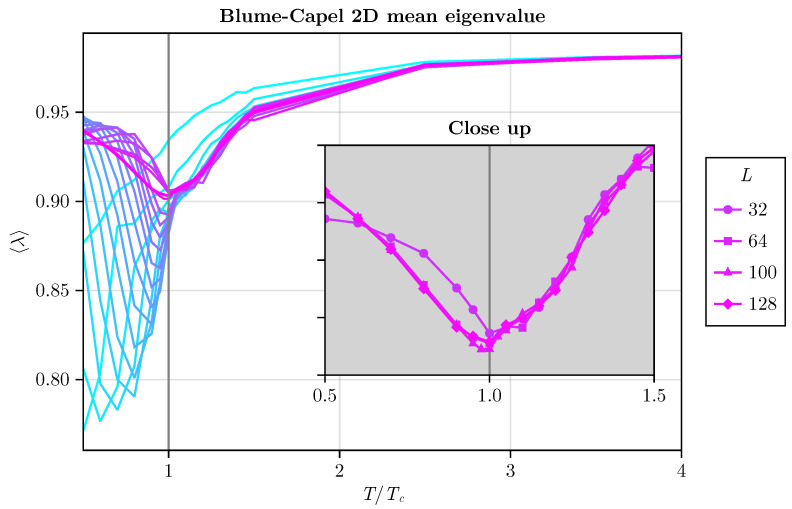
The average eigenvalue as a function of $T/T_{C}$ for different linear system sizes is depicted. We start with L=2,3,4,...,16, then proceed to larger sizes, including L=20, 25, 30, 32, 64, 100, and 128 for the two-dimensional BC model with D=0, chosen for simplicity. The inset plot illustrates that for L≥32, the minimum at $T=T_{C}$ coincides. With L≥64, there is excellent agreement.

**Figure 14 entropy-26-00395-f014:**
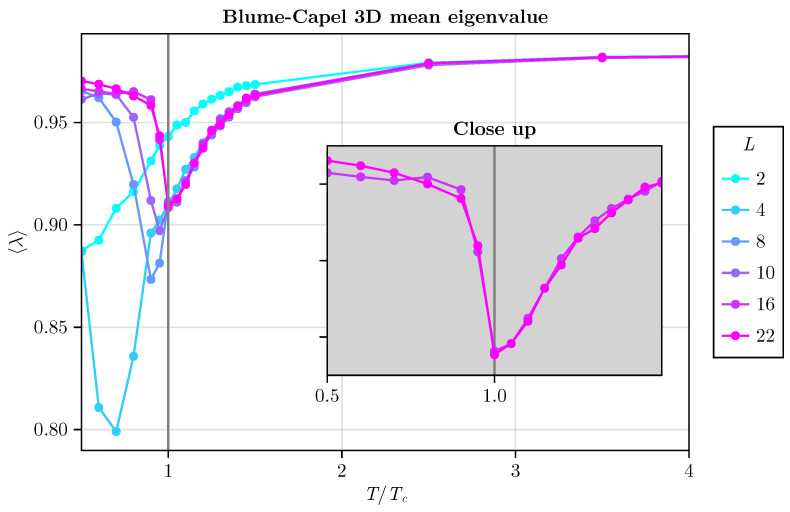
Average eigenvalue plotted against $T/T_{C}$ for various linear system sizes. We consider L=2, 4, 8, 10, 16, and 22 in a three-dimensional BC model with D=0 for simplicity. The inset plot highlights that, for L≥16, the minimum occurs at precisely $T=T_{C}$.

## Data Availability

No new data were created or analyzed in this study. Data sharing is not applicable to this article.

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
