# Peer review of "A Spectral Investigation of Criticality and Crossover Effects in Two and Three Dimensions: Short Timescales with Small Systems in Minute Random Matrices"

_entropy, 2024, doi:10.3390/e26050395_

Round 1

Reviewer 1 Report

Comments and Suggestions for Authors

This paper explores the applicability of random matrix theory in investigating critical phenomena in spin systems. Through an analysis of eigenvalue mean, dispersion, and extrema statistics, the authors demonstrate the efficacy of spectral techniques in characterizing critical points in two and three dimensions. The paper is also well-written and easy to follow. Although the theoretic analysis in the paper is rather straightforward, the idea of the paper may deserve a publication.

Author Response

We appreciate your feedback. Furthermore, we have conducted a thorough revision of our manuscript to further enhance its quality.

Reviewer 2 Report

Comments and Suggestions for Authors

This manuscript reports the application of random matrix theory, specifically focusing on Wishart-type matrices, to study critical phenomena and crossover effects in the Blume-Capel model in two and three dimensions. Through the analysis of eigenvalue statistics of a Wishart matrix, including mean, dispersion and extrema statistics, the authors demonstrate the efficacy of the spectral approach in characterizing critical points. Their central result lies in the utilization of short time scales and small system sizes, offering some computational advantages. The findings suggest that spectral methods are promising tools for elucidating the thermodynamics of complex spin systems, particularly in regimes where traditional Monte Carlo simulations face challenges due to critical slowing down and the need for large lattices. The numerical results are probably valid, but I do have a list of criticisms that should be considered by the authors before the manuscript can be considered for publication.

1)    The area of phase transitions in random matrix theory is an old one and has been subject of various reviews, see for instance [Phase Transition, Giovanni Cicuta and Luca Molinari in “The Oxford Handbook of Random Matrix Theory”]. This fact must be properly acknowledged in the introduction with a brief story of the field, citing all the relevant references.

2)    There appears to be no real progress in the understanding of the formalism in comparison with the two previous publications. The manuscript's analysis of the Blume-Capel model using Wishart-like matrices appears isolated from the broader context of phase transitions in random matrix theory. The authors fail to draw connections to existing results with polynomial potentials, despite the similarities in the observed phenomena, such as the emergence of multicut solutions and the behavior of eigenvalue densities near critical points. Clearly, there has to be an effective potential, which the authors must obtain, whose parameters drive the transition, making the zeros of the average density move continuously, and eventually two of them collide on the real axis providing mechanisms for phase transitions.

3)  The manuscript does not delve deeper into the nature of the universality class of the observed transitions or analyze the behavior of critical exponents from the perspective of the random matrix theory approach. The order of the multicriticality of the transition, for instance, could have been obtained directly from the behavior of the density near the eventual double scaling point. Exploring these aspects would provide a more comprehensive understanding of the critical behavior of the Blume-Capel model from the perspective of random matrix theory.

I summarize by saying that the manuscript reports apparently valid results, but the authors fail to make contact with the existing literature of random matrix theory on the subject. They need to address all the criticisms before it can be considered sufficient for publication.

Author Response

.
